# Tumor-Associated Neutrophils Dampen Adaptive Immunity and Promote Cutaneous Squamous Cell Carcinoma Development

**DOI:** 10.3390/cancers12071860

**Published:** 2020-07-10

**Authors:** Sokchea Khou, Alexandra Popa, Carmelo Luci, Franck Bihl, Aida Meghraoui-Kheddar, Pierre Bourdely, Emie Salavagione, Estelle Cosson, Alain Rubod, Julie Cazareth, Pascal Barbry, Bernard Mari, Roger Rezzonico, Fabienne Anjuère, Veronique M. Braud

**Affiliations:** 1Institut de Pharmacologie Moléculaire et Cellulaire, Centre National de la Recherche Scientifique, Université Côte d’Azur, UMR7275, 06560 Valbonne, Sophia Antipolis, France; khou@ohsu.edu (S.K.); apopa@cemm.at (A.P.); Carmelo.Luci@unice.fr (C.L.); bihl@ipmc.cnrs.fr (F.B.); meghraoui@ipmc.cnrs.fr (A.M.-K.); pierre.bourdely@kcl.ac.uk (P.B.); emiesalavagione@gmail.com (E.S.); cosson@ipmc.cnrs.fr (E.C.); a.rubod@laposte.net (A.R.); cazareth@ipmc.cnrs.fr (J.C.); barbry@ipmc.cnrs.fr (P.B.); mari@ipmc.cnrs.fr (B.M.); rezzonico@ipmc.cnrs.fr (R.R.); anjuere@ipmc.cnrs.fr (F.A.); 2CeMM Research Center for Molecular Medicine of the Austrian Academy of Sciences, 1090 Vienna, Austria; 3C3M, INSERM U1065, Université Côte d’Azur, 06204 Nice, France; 4Laboratory of Phagocyte Immunobiology, Peter Gorer Department of Immunobiology, Centre for Inflammation Biology and Cancer Immunology, King’s College London, London SE1 1UL, UK

**Keywords:** neutrophils, cutaneous squamous cell carcinoma, PD-1, PD-L1, gene expression profile

## Abstract

Cutaneous squamous cell carcinoma (cSCC) development has been linked to immune dysfunctions but the mechanisms are still unclear. Here, we report a progressive infiltration of tumor-associated neutrophils (TANs) in precancerous and established cSCC lesions from chemically induced skin carcinogenesis. Comparative in-depth gene expression analyses identified a predominant protumor gene expression signature of TANs in lesions compared to their respective surrounding skin. In addition, in vivo depletion of neutrophils delayed tumor growth and significantly increased the frequency of proliferating IFN-γ (interferon-γ)-producing CD8+ T cells. Mechanisms that limited antitumor responses involved high arginase activity, production of reactive oxygen species (ROS) and nitrite (NO), and the expression of programmed death-ligand 1 (PD-L1) on TAN, concomitantly with an induction of PD-1 on CD8^+^ T cells, which correlated with tumor size. Our data highlight the relevance of targeting neutrophils and PD-L1-PD-1 (programmed death-1) interaction in the treatment of cSCC.

## 1. Introduction

Cutaneous squamous cell carcinoma (cSCC) is the second most common non-melanoma skin cancer, which is associated with alterations in immunity that favor inflammation and tumor development [1]. cSCCs are generally cured with surgery, but they can reach a stage of advanced invasive disease associated with extremely rapid local relapse, for which no efficient therapy has been approved so far. Antitumor activities of immune cells have been shown to be central to immune surveillance of the skin. Indeed, immunosuppressed organ transplant patients with defective T cell responses display a high incidence of cSCC [2] and high numbers of regulatory T cells are detected in these tumors [3]. In addition, when CD8+ T cells, γδ T cells, and natural killer (NK) cells are rendered anergic by downregulation of NKG2D induced by high expression of its ligands on tumors, increased cancer incidence is observed [4]. The reduction of DMBA/PMA (7,12-dimethylbenz[a]anthracene /phorbol 12-myristate 13-acetate) skin carcinogenesis in CXCR2−/− (C-X-C motif chemokine receptor 2) mice also suggests that myeloid suppressor cells play a role, but their full characterization has not been done yet [5]. This is consistent with high-risk cSCC patients harboring an increased number of circulating and tumor-resident neutrophils [6].

Neutrophils are highly abundant immune cells in the tumor microenvironment (TME) of a large number of cancers. They seem to regulate the initiation and progression of cancer, but their role is still a matter of controversy. In vivo depletion experiments led to either a failure to control tumor growth [7,8] or a reduction of tumor progression [9]. Both antitumor and protumor functions have been assigned to tumor-associated neutrophils (TANs), with this functional plasticity being regulated by factors of the TME [10,11,12]. TGF-β (Transforming growth factor beta) promotes protumor (N2) TAN while IFN-β and a lack of TGF-β favor antitumor (N1) TAN [9,13]. In humans, a meta-analysis of gene expression signatures among 39 human malignancies revealed that within the leukocyte population in the tumor microenvironment, polymorphonuclear (PMN) cells have been linked to the most adverse prognosis [14]. The blood neutrophil-to-lymphocyte ratio (NLR) has also been proposed as a prognostic factor, with a high NLR associated with poor survival [15]. However, a consensus on the classification of neutrophil subsets is lacking. While the capacity to kill or inhibit the growth of tumor cells is endorsed by N1 TAN, promotion of extracellular matrix remodeling, angiogenesis, cancer cell invasion, metastasis, and immune suppression are attributed to N2 TAN, also named PMN-myeloid-derived suppressor cells (PMN-MDSCs) [16,17]. Their phenotype largely overlaps and may even represent a continuum of states rather than clear distinct subsets [18].

Because high-risk cSCC in patients is associated with an increased number of circulating and tumor-resident neutrophils [6], we undertook a study to fully characterize TAN’s phenotypes and functional roles in cSCC. A key issue to consider is the plasticity of cells in various microenvironments. This is particularly true for neutrophils, as their transcriptomic profiles have been found to be highly divergent when analyzed in the blood, bone marrow (BM), spleen, and tumor [19,20,21]. We therefore compared neutrophils isolated from precancerous lesions or established cSCC induced by a DMBA/PMA treatment [22] and from the orthotopic implantation of a DMBA/PMA-derived SCC cell line, mSCC38, in the skin dermis [23], with neutrophils isolated from the skin surrounding these tumors. We show that the majority of TANs within precancerous lesions and cSCC display protumor functions contrary to neutrophils isolated from the surrounding skin. We also define key features of protumor TANs in cSCC that highlight the functional relevance of targeting neutrophils in this cancer.

## 2. Results

### 2.1. The Local Microenvironment of Cutaneous Squamous Cell Carcinoma Imprints Tumor-Associated Neutrophils Towards a Protumor Phenotype

Application of DMBA together with PMA on mouse skin is a widely used experimental model to study skin carcinogenesis [24]. This model accurately mimics the different stages of skin carcinoma development in humans, by inducing papilloma, which can further develop into invasive cSCC upon PMA stimulation (Appendix A). To assess the role of TAN in cSCC, we focused our study on Gr-1^bright^/Ly6G^+^ neutrophils that extravagated into tissue, either within the skin surrounding precancerous lesions (papilloma skin) and established carcinomas (tumor skin), or those infiltrating the precancerous lesions (papilloma) and carcinomas (tumor). We detected a significant and progressive increase of the proportion of neutrophils within papillomas and tumors compared to the surrounding skin, with TAN accounting for 30–80% of tumor-infiltrating CD45^+^ cells (Figure 1A). We purified these neutrophils by flow cytometry (Appendix A) and performed unbiased gene expression array analysis to investigate their molecular signatures throughout cSCC development. These cell-sorted Gr-1^bright^ cells exclusively expressed Ly6G, indicating they were neutrophils and not monocytes (Appendix A) [25]. In addition, we excluded eosinophil contamination as eosinophil peroxidase encoded by *Epx* (eosinophil peroxidase) was not found to be expressed in the gene signature (Appendix A) [26]. Principal component analysis (PCA) of the 15% most variable probe sets revealed a high reproducibility across replicates and a clear segregation according to biological condition (Figure 1B). The first principal component axis (PC1), the one explaining the largest percentage of variability (69.05%), separated the surrounding skin from cancer lesion samples. The identity of papilloma versus tumor stages was illustrated by the PC2, explaining 15.76% of the variability. The unique gene expression signature of each cell origin was further highlighted in the hierarchical clustering (Appendix A).

We next examined the differentially expressed genes (DEGs) across pairwise comparisons (Figure 1C). The most divergent contrast was the comparison of the tumor with its control tumor skin, yielding 2045 DEGs, followed by the comparison of papilloma with control papilloma skin, yielding 1528 DEGs. In these two contrasts, a majority of DEGs were upregulated. To a lesser extent, we observed 235 and 144 differentially modulated genes for the contrasts of tumor versus papilloma and tumor skin versus papilloma skin, respectively. In order to compare the modulation between lesions versus skin controls, across the papilloma and tumor stages, we performed the genuine association of expression profiles (GENAS) method from the limma R package. We compared the log fold changes between these two most divergent contrasts on the totality of probe sets, without setting differential expression cutoffs (Appendix A). The GENAS analysis yielded a strong biological correlation coefficient of 0.76 (*p* < 0.001), indicating that TAN gene modulation onset takes place at the papilloma stage and follows a similar trend in the tumor.

Then, to apprehend the biological processes and molecular functions associated with TANs in the context of cutaneous carcinoma pathogenesis, we performed a core analysis of the DEGs we obtained using ingenuity pathways analysis (IPA). As shown in Figure 1D, TGF-β, TNF-α, and IFN-γ were the most significantly activated upstream regulators in lesions compared to the surrounding skin. These cytokines have been associated with protumorigenic phenotypes [9,27,28,29], suggesting that the cSCC microenvironment favors the tumor-promoting TAN phenotype. Consistent with this observation, the analysis of the canonical pathways enriched and modulated in at least one of the three contrasts revealed that CXCL-8 (IL-8) signaling was the most significantly activated pathway (Appendix A). This analysis also revealed that signaling through CXCR1 and CXCR2 receptors expressed by TANs resulted in the activation of specific functions promoting tumor growth, such as angiogenesis, endothelial cell migration, tumor invasion, and inflammation, but also in the inhibition of apoptosis, suggesting they harbor an increased lifespan in lesions (Appendix A). In mice, CXCL1 (C-X-C motif ligand 1) (KC) and CXCL2 (MIP-2, CXCL5 (LIX)) have been described as the functional homologues of human CXCL8 (IL-8) and they bind to CXCR1 and CXCR2 receptors, which we found to be expressed on mouse neutrophils (Appendix A). Neutrophil adhesion and chemotaxis were also activated, consistent with the well-known role of CXCL8 (IL-8) in the recruitment of neutrophils to the tumor site and in correlation with the increased proportion of TANs found in lesions compared to skin controls (Figure 1A). Altogether, these data highlighted the enhanced protumor functions of TAN in lesions compared to the surrounding skin.

We then clustered (K-means, ExpressCluster) the union of the differentially modulated genes across biological conditions (2166 probe sets) in order to identify specific gene expression signatures. We obtained 10 distinct clusters associated with a specific pattern of gene expression, revealing a clear separation between skin controls and lesions (Figure 1E). We further performed a Protein Analysis Through Evolutionary (PANTHER) enrichment analysis on the genes in each of the 10 clusters (Appendix A). With this approach, we could retrieve both shared and specific functions of TAN and link them to localization in skin or lesions and to stages of carcinogenesis. The GO (gene ontology) enrichments revealed that the main features of TAN once infiltrated into papillomas were more specifically related to extracellular matrix remodeling (cluster 1 and 7), angiogenesis, and metastasis (cluster 7). When infiltrated into tumors, TANs were associated with active glycolysis (clusters 5, 8, and 9), increased survival, and immune suppression (cluster 10). TANs within papillomas and tumors also shared features related to repressed cytoskeleton reorganization controlling neutrophil functions (cluster 2), repressed leukocyte extravasation and migration (cluster 3), decrease of specific inflammatory and immune responses (cluster 3 and 4), and response to CXCL8 (IL-8) and TGF-β (cluster 6 and 7). Overall, a dominant protumor gene expression profile was observed. This can be illustrated by the significant upregulation of genes previously linked to protumorigenic phenotypes: *Siglec5* (encoding for Siglec F) in cluster 5, *Cd274* (encoding for PD-L1), *Vegfa, Olr1* (encoding for LOX-1) in cluster 6, *Nos2* in cluster 9, and *Arg1* in cluster 10 (Figure 1F) [30,31,32,33,34]. This gene signature was also found to be significantly shared with the gene signature of protumor neutrophils infiltrating lung SCC [35] (Appendix A). Pairwise comparisons of DEGs from our study with DEGs from TAN in lung SCC compared to circulating neutrophils [35] identified a positive correlation with the contrasts of papillomas and tumors over surrounding skin, revealing that TANs from lung and cutaneous SCC share gene expression profiles (Appendix A).

Altogether, these analyses identify a massive infiltration of TANs in cSCC lesions, which display a dominant TAN’s protumor gene expression profile that differ from their skin counterparts, highlighting a strong impact of the local microenvironment.

### 2.2. TAN Exhibit Heterogeneity in Their Phenotype and Exert Protumor Functions

To further characterize TAN in invasive cSCC, we took advantage of a recently established cSCC cell line mSCC38, which was generated from DMBA/PMA-treated mice [23]. Orthotopic implantation of tumorigenic mSCC38 in the skin dermis promoted tumor growth in 100% of mice, over the course of ~30 days (Figure 2A), mimicking human invasive cSCC with invasion in the dermis (Figure 2B). Similar to the DMBA/PMA cSCC mouse model, the frequency of neutrophils increased in the grafted mSCC38 tumor over time (Figure 2C) and their proportion among infiltrating CD45+ immune cells was positively correlated with tumor volume (Figure 2D). Immunofluorescence staining localized neutrophils within the tumor bed (Figure 2E). Simultaneously, the frequency of neutrophils increased in the blood while no modulation was observed in the skin, spleen, and BM (Appendix A).

We used this mSCC38 mouse model to further investigate the phenotypic and functional diversity of TANs in cSCC. First, we evaluated the co-expression of a selected set of activation markers and cell surface receptors on live CD45+Ly6G+ neutrophils at a single-cell level by flow cytometry. t-distributed stochastic neighbor embedding (tSNE) analysis was used to create a single common map of neutrophils across all samples (bone marrow, spleen, blood, skin, and tumor) using markers identified in the above transcriptomic data (PD-L1 (CD274) and Siglec F) (Figure 1) and previously described as being modulated in neutrophils (CD54, CD62L, CD11b, CD80, CD11c) [32,34]. We could not include LOX1 because of a lack of available antibody. In an unsupervised manner, the t-SNE analysis arranged the bone marrow neutrophils in the lower left area of the map whereas spleen and blood neutrophils were in the central right and central left areas, respectively. The skin and tumor neutrophils shared the upper right area, with an additional specific location for tumor neutrophils in the upper left area (Figure 2F). This approach revealed the heterogeneity of neutrophils in tumors (TANs) compared to skin but also some similarities when both were compared to neutrophils from the spleen, blood, and bone marrow (Figure 2F). Individual expression of each marker per organ also reflected such heterogeneity (Appendix A). To identify the phenotype of TANs, we used the FlowSOM clustering tool to separate neutrophil subsets into 49 meta-clusters (MCs) (Figure 2F). The complete linkage hierarchical clustering of both samples and mean-centered MC cell proportions revealed four MCs highly abundant in tumor samples (Figure 2G, green gates); here, two of them were composed of PD-L1+ CD54^+^ Siglec F+ neutrophils (MC11 and MC17) and the two others contained PD-L1+ CD54+ Siglec F^−^ neutrophils (MC31 and MC13) (Figure 2H), with mean proportions of 16.68 ± 3.06% and 17.41 ± 2.49%, respectively, in tumor samples (Figure 2I).

This computational analysis of the neutrophil phenotype revealed the PD-L1 marker as highly discriminant of TAN. Siglec F^+^ neutrophils were previously found to accumulate in lung cancer, exerting protumor functions [32]. By manual gating on cSCC neutrophil subsets, we found elevated proportions of TAN expressing both (14%) or one of the two markers, PD-L1 (15%) and Siglec F (15%), compared to skin with only 9% of PD-L1^−^ Siglec F^+^ neutrophils. Spleen and BM neutrophils were negative for these markers (Figure 3A). Importantly, we found tumor size to correlate positively with the frequency of PD-L1^+^ TAN, independently of Siglec F expression, and negatively with PD-L1^−^ Siglec F^−^ TAN (Figure 3B). These findings indicate that TAN can share both activation (CD54) and protumor markers (PD-L1, Siglec F) [32,34,36] and that PD-L1 expression on TAN is associated with cSCC growth.

We further investigated whether TAN displayed functional properties contributing to tumor progression. Reactive oxygen and nitrite species are produced by neutrophils in various conditions of activation and are known to participate in tumor progression [37]. The intracellular levels of the reduced form of nicotinamide adenine dinucleotide (NAD) and nicotinamide adenine dinucleotide phosphate (NADP) are linked to the production of ROS. Using a fluorescent sensor, we measured a significant increase of intracellular levels of NAD(P)H (reduced nicotinamide adenine dinucleotide phosphate) in neutrophils infiltrating tumors compared to skin (Figure 3C). Spleen neutrophils behaved like TAN while BM neutrophils had NAD(P)H levels similar to skin neutrophils. Complementary to these results, we found an elevated production of ROS in TANs, lower levels in the spleen, and poor ROS production in the skin and BM (Figure 3D). The neutrophil subsets expressing PD-L1 also produced significantly more ROS than the subsets lacking PD-L1 (Figure 3E). Besides, we quantified the nitrites produced by highly purified neutrophils from tumors of mSCC38-bearing mice. Because of the paucity of neutrophils in the skin, we were not able to include the skin control, but we added neutrophils from the BM instead. As shown in Figure 3F, purified TANs produced significantly more nitrite (NO). Similarly, we quantified arginase activity in highly purified neutrophils and found significant increased activity of arginase in TANs (Figure 3G). Elevated NO and arginase activity in the mSCC38 model was consistent with our finding that transcription of *Nos2* and *Arg1* was increased in neutrophils from tumors and papillomas of DMBA/PMA-treated mice, compared to surrounding skin (Figure 1F). Such a phenotype also suggests that TANs may be involved in T cell immunosuppression in cSCC.

### 2.3. TANs Limit Antitumor CD8^+^ T Cell Responses and Concomitant Upregulation of PD-L1 on TANs and PD-1 on CD8+ T Cells Participates in This Process

The above gene expression analysis and phenotypic and functional characterization suggested that TANs could promote cSCC. Consistent with this phenotype, we detected TGF-β within whole tumor cell lysates, with a 2-fold increase of TGF-β between days 15 and 28 (Appendix A). In order to directly demonstrate such a protumor role of TANs, we developed an in vivo protocol of neutrophil depletion that allowed the depletion of neutrophils in blood and in tumors (Appendix A). As shown in Figure 4A, the depletion of TANs significantly delayed mSCC38 growth as compared to isotype control IgG-treated mice.

Immune suppression is associated with protumor functions and has been attributed to N2 TAN and MDSC [10,16]. We therefore evaluated whether TAN promoted cSCC progression through the inhibition of antitumor CD8^+^ T cell responses. We chose to address this question by directly monitoring ex vivo the outcome of TAN depletion on the CD8^+^ T cell response. This choice of approach was guided by recent concerns over the accuracy of the in vitro assays used to measure suppression of T cell activity [38]. The immunosuppressive activity of TAN was observed when comparing the CD8^+^ T cell responses in tumors from mice that were treated either with anti-Ly6G or isotype IgG control (Figure 4B,C).

We did not find significant differences in the frequencies of CD3+CD8+ and CD3+CD4+ T cells, as well as Foxp3^+^ CD25^+^ regulatory T cells (Treg) and natural killer (NK) cells, among mSCC38-infiltrating CD45^+^ immune cells (Appendix A). By contrast, the frequencies of proliferating and IFN-γ-producing CD3^+^CD8^+^ T cells were significantly increased in the absence of neutrophils (Figure 4B,C). This immune suppression may result from a direct crosstalk between TAN and CD3+CD8+ T cells, as immunofluorescence staining of mSCC38 tumor sections identified neutrophils in contact with CD8+ T cells (Figure 4D).

Several mechanisms can be responsible for the inhibition of CD3+CD8+ T cell responses. We found that TAN showed a high arginase activity (Figure 3G) and high production of ROS and NO (Figure 3C–F) that can limit CD8+ T cell responses. In addition, because we found PD-L1 to discriminate TANs from neutrophils of the skin, we investigated whether PD-L1-PD-1 interaction could also play a role. We analyzed the expression of PD-1 on T cells (Figure 4E) and detected highly significant increased frequencies of PD-1-expressing CD8+ and CD4+ T cells in the tumor (*p* < 0.0001). By contrast, no expression of PD-1 was detected on T cells from the spleen and lymph node of tumor-bearing mice. To further depict mechanisms that could sustain such a role, we investigated the effect of the tumor microenvironment on the induction of PD-L1 expression and on TAN survival. We prepared tumor-conditioned medium (TCM) and treated in vitro PD-L1-negative BM neutrophils from naive mice. TCM was able to induce PD-L1 on about 40% of neutrophils (Figure 4F and Appendix A). We then tested the impact of TGF-β, TNF-α, and IFN-γ cytokines as we previously identified them as top upstream regulators in TAN from DMBA/PMA-treated mice and also GM-CSF (granulocyte-macrophage colony-stimulating factor), which was reported to induce PD-L1 [39]. Treatment with IFN-γ, TNF-α, and GM-CSF could induce PD-L1 with different efficiencies, with IFN-γ being the most efficient and GM-CSF the least (Appendix A). TGF-β did not induce any PD-L1 expression on neutrophils. We confirmed the predominant role of IFN-γ by depleting the TCM of IFN-γ and found that the frequency of PD-L1+ neutrophils was decreased by half (Figure 4F). The TCM was also able to maintain neutrophils alive in this experimental setting (Appendix A), similarly to GM-CSF. This was in agreement with our previous observation from the gene expression profile of TANs from DMBA/PMA-treated mice, which showed that TANs displayed an increased lifespan compared to neutrophils in the surrounding skin (Appendix A).

Collectively, our data demonstrate that infiltrated TANs contribute to cSCC development by limiting effector CD8^+^ T cell responses.

## 3. Discussion

The observed increased infiltration of neutrophils in invasive cSCC in humans [6] questions their role during cSCC development and relapse. To gain an understanding of their contribution, we provide here an extensive characterization of neutrophils infiltrating lesions and the surrounding skin throughout cSCC progression. Both gene expression signatures, and phenotypic and functional analyses showed that, once within lesions, TANs acquired protumorigenic phenotypes and immunosuppressive capacities that contribute to cSCC progression.

The recruitment of abundant numbers of neutrophils was observed in the mouse models of cSCC we studied, consistent with the human pathology [6]. This may be a feature shared with squamous cell carcinomas in general, as depicted for lung SCCs enriched in TANs as compared to lung adenocarcinomas [35,40,41]. Interestingly, we found a significant correlation between gene signatures of TAN from both cutaneous and lung SCC, suggesting that the immune contexture of SCC certainly favors neutrophil recruitment. This is illustrated by our finding that the top canonical pathway that was activated was the IL-8 signaling pathway. This confirmed a primary role of CXCR1/CXCR2 and their ligands in the recruitment of neutrophils from blood, in agreement with other cancer types [5,17,42].

To study the impact of the TME, we chose to compare TANs to neutrophils within the skin surrounding the lesions. We speculated that as they were both recruited from blood, they were the most accurate controls. The study of the papilloma stage, which can be viewed as a precancerous step, together with the tumor stage allowed us to show that most of the impact of the TME already occurred at the papilloma stage. Nevertheless, we were also able to identify specific features of TANs at each stage of the carcinogenesis. We found that angiogenesis and extracellular matrix remodeling were functions primarily implemented by precancerous TANs and that immune suppression was predominantly completed at the tumor stage. This indicates that neutrophils are highly plastic, not only in distinct organs [19,20,43] but also over the course of tumor progression. The comparison of TME versus surrounding skin also identified a prolonged lifespan of TAN within tumors, and in vitro cultures in the presence of TCM enhanced the survival of neutrophils. This sustained survival with TCM was also observed in other cancers [39,40] and we identified a role for GM-CSF in this process. This observation is reminiscent of the reduced rate of apoptosis of the circulating human low-density neutrophils (LDNs) compared to high-density neutrophils (HDNs) and reinforced the proposed link between LDNs and N2 TANs [30,44].

The observation that TANs predominantly harbor a protumorigenic phenotype in cSCC is shared with many cancers but also differs from others [11,14,16]. Our study does not exclude the presence of minor subsets of neutrophils that favor antitumor responses, as reported in the early stages of lung cancer [45] or in a murine carcinogen-induced sarcoma model [8]. Hence, we detected higher levels of *Nos2* and *GrzB* transcripts in TANs, with both being associated with cytotoxic functions [46,47,48]. We also found that TANs were heterogeneous in cSCC, with upregulation of some activation markers. Whether anti- and pro-tumor neutrophils could be differentiated by the level of activation still remains an open question [9,36]. In human studies, antitumor TANs have been shown to express a classically activated phenotype with upregulation of CD54 and downregulation of CD62L and CD16 [40,49] and protumor TANs also showed an activated immunosuppressive phenotype with upregulation of CD54 and PD-L1, similarly to our findings [39]. IFN-γ drove a high level of PD-L1, consistent with an adaptive resistance [50,51]. Further work is needed to get the full diversity of phenotypes and their relationships with PMN-MDSC and immature/mature neutrophils [52,53]. Already, comparison with lung SCC models suggests that conserved gene expression profiles can be found and that a continuum of states also characterizes neutrophil subsets [18,35].

A key observation in our study is that TANs in cSCC are actively involved in immune suppression, limiting CD8^+^ T cell responses, and promoting cSCC growth. We investigated the immunosuppressive phenotype of TANs in vivo, in the complex context of tumor progression and immune response modulation. We demonstrated that a combination of immunosuppressive mechanisms was in place. TANs purified from cSCC tumors exhibited increased *Arg1* transcription as well as higher arginase activity, potentially leading to arginine depletion from the environment, a metabolite crucial for T cell effector function [11,16]. At the same time, we showed an increase of NO and ROS, both being able to act as intracellular signaling molecules to modulate T cell functions [54]. Their contribution is likely to be local and hence, we did not find significant modulation of nitrite concentrations and arginase activity in whole tumor cell samples from mice depleted or not of neutrophils. Further work is needed to assess the full contribution of NO and ROS. Lastly, we found that TANs upregulated PD-L1 while CD8^+^ T cells upregulated PD-1 in the TME. TANs can participate in the immune suppression of T cell responses via PD-L1-PD-1 immune checkpoint interaction, as indicated by the positive correlation between tumor size and frequencies of PD-L1+ TANs. This is consistent with recent studies, which demonstrated the contribution of both tumors and non-tumor cells expressing PD-L1 to the suppression of T cell responses [55]. Interestingly, a recent phase 1 trial using cemiplimab that targets PD-1 was conducted in advanced cSCC. It induced a response in half of the patients and may be linked to PD-L1+ TAN infiltration [56]. PD-L1 expression on TANs was reported in several cancers, such as head and neck squamous cell carcinoma, hepatocellular carcinoma, colon cancer, gastric cancer, and lung cancer, and may constitute targets for anti-immune checkpoint treatments [55]. The effect of current anticancer immunotherapy treatments in combination with other molecules acting on TANs is also being examined and may provide valuable insights into TAN functions. This is well exemplified by Gemcitabine treatment, which selectively eliminated CD11b^+^ Gr-1^+^ cells and efficiently enhanced the efficacy of a combination of resiquimod immunomodulatory and PD-L1 blockade against a murine HNSCC tumor highly infiltrated by TANs [57], or by c-MET (tyrosine-protein kinase Met) inhibitors, which enhanced anti-PD-1 treatment efficacy via blockade of neutrophil recruitment [27]. Besides PD-L1, we evaluated the expression of Siglec F reported to be a protumor marker for TANs in lung adenocarcinomas [32]. Siglec F was induced on a proportion of TANs in cSCC, either together with PD-L1 or alone, but its expression was not associated with ROS, nor correlated with tumor size in cSCC. Further characterization of TAN subsets is warranted to extend our understanding of their functions in cSCC development.

## 4. Materials and Methods

### 4.1. Mice

FVB/N wild-type (WT) mice (Charles River Laboratories, St Germain Nuelles, France) were bred and housed in specific pathogen-free conditions. Experiments were performed using 6–7-week-old female FVB/N, in compliance with institutional guidelines and were approved by the regional committee for animal experimentation (reference MESR 2016112515599520; CIEPAL, Nice Côte d’Azur, France).

### 4.2. In Vivo Tumor Growth

Multi-stage chemical carcinogenesis was induced as follows: 200 nmol of DMBA (Sigma-Aldrich, St Quentin Fallavier, France) dissolved in toluene (Merck Millipore, Molsheim, France) and further diluted in acetone were applied on the shaved back of mice at 6 and 13 weeks of age, together with the application twice weekly of 5 nmol of PMA (Sigma-Aldrich, St Quentin Fallavier, France) dissolved in acetone, from week 1 to week 18–20. Mice were assessed for papilloma and tumor development throughout the treatment from week 10–12 up to sacrifice in week 22. The mSCC38 tumor cell line was established from DMBA/PMA-induced sSCCs and maintained in DMEM (Dulbecco’s Modified Eagle Medium) (Gibco-ThermoFisher Scientific, Courtaboeuf, France) supplemented with 10% heat-inactivated fetal bovine serum (FBS) (GE Healthcare, Chicago, IL, USA), penicillin (100 U/mL), and streptomycin (100 μg/mL) (Gibco-ThermoFisher Scientific, Courtaboeuf, France). Then, 5 × 10^5^ mSCC38 were intradermally injected in anesthetized mice after dorsal skin shaving. Tumor volume was measured manually using a ruler and calculated according to the ellipsoid formula: Volume = Length (mm) × Width (mm) × Height (mm) × (π/6).

### 4.3. In Vivo Neutrophil Depletion

Mice received intraperitoneal injection of 150 μg of either anti-Ly6G (clone 1A8) or isotype IgG control (clone 2A3) (BioXcell, Lebanon, NH, USA) one day before mSCC38 engraftment and continuously every three days. Neutrophil depletion was monitored by flow cytometry, in blood day 14 and 28, and in tumor at sacrifice.

### 4.4. Tissue Preparation and Cell Purification

Papillomas and tumors were enzymatically treated twice with collagenase IV (0.6 mg/mL) (Sigma-Aldrich, St Quentin Fallavier, France), dispase II (2.5 mg/mL), and DNase I (0.2 mg/mL) (Roche Diagnostic, Meylan, France) for 20 min at 37 °C and the skin surrounding cancerous lesions was treated twice with collagenase IV (0.6 mg/mL) for 20 min at 37 °C. Immune cells were enriched using a Percoll gradient centrifugation (GE Healthcare, Chicago, IL, USA). For DMBA/PMA-treated mice, isolated cells were incubated with 5 μg/mL anti-CD16/CD32 (2.4G2) to block Fc receptors prior to incubation with the following fluorescently labeled antibodies to CD3 (145-2C11), CD11c (HL3), CD45 (30-F11), CD45R/B220 (RA3-6B2), CD335/NKp46 (29A1.4), Gr-1 (RB6-8C5), Ly6C (AL-21), Ly6G (1A8), and MHCII (I-A/I-E) (2G9). For mSCC-38-bearing mice, isolated cells were enriched in CD45^+^ cells using CD45-biotin antibody (BD Biosciences, Le Pont de Claix, France) and anti-biotin magnetic beads (Miltenyi Biotec, Paris, France) according to the manufacturer’s instructions. Cells were then incubated with 5 μg/mL anti-CD16/CD32 (2.4G2) and stained with anti-CD11b (M1/70), anti-Ly6G (clone 1A8) antibodies, and live/dead marker (7-AAD). All antibodies and viability markers were purchased from BD Biosciences, Le Pont de Claix, France. Neutrophils were cell sorted on a BD FACSAria II™ (BD Biosciences, Le Pont de Claix, France). Purity after cell sorting was >98%.

Spleen and bone marrow (BM) cells flushed from mice femurs were homogenized into single-cell suspensions through a 70-μm nylon mesh. Blood was collected from the tail vein on heparin tubes. Red blood cells were lysed in ACK lysis buffer (150 mM NH4Cl, 10 mM KHCO3, 0.1 mM Na2 EDTA, pH 7.2). Neutrophils were enriched from BM cell suspensions using Histopaque-based density gradient centrifugation (Sigma Aldrich, St Quentin Fallavier, France).

### 4.5. RNA Isolation and Microarray Analysis

First, 50 to 700 ng total RNA were isolated from 0.01 to 2 million sorted cells using a micro RNAeasy kit according to the manufacturer’s instructions (Qiagen, Courtaboeuf, France). Purity and quality were assessed with a Bioanalyser (Agilent Technologies, Les Ulis, France). Probes were synthesized from RNA with the LowInput QuickAmp Labeling Kit (Agilent Technologies, Les Ulis, France). cRNA with appropriate quality control were hybridized to SurePrint G3 Mouse GE 8x60K microarrays (Agilent Technologies, Les Ulis, France). Two or three biological replicates were performed for each experimental condition to obtain sufficient material. Neutrophils were purified from the skin surrounding papillomas (3 mice per experiment), skin surrounding tumors (5 mice per experiment), papillomas (a mean of 60 papillomas from 3 mice), and tumors (a mean of 12 tumors from 5 mice). The obtained microarray experimental data and associated microarray designs were deposited in the NCBI Gene Expression Omnibus (GEO) (http://www.ncbi.nlm.nih.gov/geo/) under the serial record number GSE133807. The raw data were quantile normalized using the Bioconductor package limma. The batch effect induced by the microarray chips was removed using the ComBat method. Gene expression was implemented with the limma package. The Benjamini–Hochberg procedure was used to control the experiment-wise false discovery rate (FDR) from multiple testing procedures. We performed a principal component analysis (PCA) on the 15% most variable probe sets with a minimum log2 average expression of 6, selected with the PopulationDistances program from http://cbdm.hms.harvard.edu/LabMembersPges/SD.html. Using a pairwise average-linkage method (GenePattern) (http://www.broadinstitute.org/cancer/software/genepattern/), a hierarchical clustering was performed on the same probe sets median centered across all samples. Differentially expressed probe sets were selected based on an average log2 expression level across all conditions of at least 6, an absolute log2-fold change ≥1, and an adjusted *p* value ≤ 0.05. Further analyses were carried out using the Protein Analysis Through Evolutionary (PANTHER) database (http://pantherdb.org) and QIAGEN’s Ingenuity Pathway Analysis (IPA) tool, QIAGEN Redwood City (http://www.qiagen.com/ingenuity). A biological correlation coefficient between papilloma versus papilloma skin and tumor versus tumor skin comparisons was estimated using the Genuine Association of Gene Expression Profiles (GENAS) method implemented in the limma R package. A k-means ++ (z-norm) clustering was performed with the ExpressCluster software v1.3 (http://cbdm.hms.harvard.edu/LabMembersPges/SD.html) on the union of differentially modulated genes between lesions and surrounding controls (a total of 2166 probe sets). For genes with multiple probe sets, the median intensity value was computed.

### 4.6. Flow Cytometry and Computational Analysis

Cell suspensions were incubated with anti-CD16/32 (2.4G2) to block Fc receptors. For flow cytometry surface labelling, cells were stained with antibodies against CD3 (145-2C11), CD4 (GK1.5 and RM4-5), CD8α (53-6.7), CD11b (M1/70), CD19 (1D3), CD25 (PC61), CD45 (30-F11), MHCII (M5/114.15.2), Gr1 (RB6-8C5), Ly6G (1A8), Ly6C (AL-21), NKp46 (29A1.4), Siglec F (E50-2440), CD274 (MIH5), CD62L (MEL-14), CD54 (3E2), CD80 (16-10A1), CD279 (J43) (BD Biosciences, Le Pont de Claix, France), CD11c (N418), (Biolegend, Amsterdam, The Netherlands), and live/dead markers: 7-AAD (BD Biosciences, Le Pont de Claix, France), Zombie Aqua, and Zombie NIR (Biolegend, Amsterdam, The Netherlands). For intracellular staining, cells were either stained after isolation or restimulated for 4 h at 37 °C in complete DMEM medium supplemented with 100 ng/mL PMA (Sigma-Aldrich, St Quentin Fallavier, France), 1 μg/mL ionomycin in the presence of GolgiStop, and GolgiPlug (BD Biosciences, Le Pont de Claix, France). Cells were fixed and permeabilized with either BD Cytofix/Cytoperm (BD Biosciences, Le Pont de Claix, France) for cytokine staining or Foxp3/Transcription Factor Staining Buffer Set (eBioscience, Paris, France) for nuclear staining. Antibodies against Ki67 (SolA15), IFN-γ (XMG1.2) and Foxp3 (MF23) were used. Samples were acquired on a BD LSR Fortessa (BD Biosciences, Le Pont de Claix, France) and analyzed with DIVA V8, FlowJo V10 software (BD Biosciences, Le Pont de Claix, France) and a Cytobank platform (Beckman Coulter, Roissy, France). The visualization of t-Distributed Stochastic Neighbor Embedding (viSNE implementation of t-SNE) was used to automatically arrange cells according to their expression profile of the measured proteins and to visualize all cells in a 2-D map, where the position represents local phenotypic similarity. The analyzed neutrophils were embedded in a set of t-SNE axes designated as t-SNE-1 and t-SNE-2 according to the per-cell expression of CD11b, Ly-6G, CD62L, PD-L1 (CD274), CD54, and Siglec F. After dimensionality reduction with t-SNE, neutrophils were grouped in 50 meta-clusters that contained phenotypically homogenous cells using the computationally generated self-organized map with the FlowSOM algorithm.

### 4.7. Immunofluorescence

mSCC38 tumors were fixed in Antigenfix (Diapath, Martinengo, Italy) for 1 h at 4 °C, washed, and incubated in 30% sucrose (Sigma-Aldrich, St Quentin Fallavier, France) overnight at 4 °C. Fixed tumors were washed, embedded in OCT (Tissue-Tek, Villeneuve d’Ascq, France), and frozen prior to cryostat sectioning. Next, 7-μm-thick cryostat tumor sections were blocked with 10% normal donkey serum in PBS, 2% BSA (Sigma-Aldrich, St Quentin Fallavier, France), 1% FBS, and 0.5% saponin (Sigma-Aldrich, St Quentin Fallavier, France) at room temperature (RT) for 1 h. Sections were stained overnight with primary antibody (purified anti-CD8 (53-6.7) or purified anti-CD31 (390) (BD Biosciences, Le Pont de Claix, France), followed by incubation for 2 h at RT with donkey anti-rat IgG-A594 (Invitrogen-ThermoFisher Scientific, Courtaboeuf, France). After extensive washes, sections were labeled with anti-Ly6G-FITC (1A8) for 1 h at RT. Nuclei staining was performed using Hoechst 33342 (ThermoFisher Scientific, Courtaboeuf, France) for 5 min at RT and sections were mounted with Prolong Diamond (ThermoFisher Scientific, Courtaboeuf, France). Slides were imaged with a LSM780 confocal microscope (Zeiss, Marly le Roi, France) and analyzed with Fiji software.

### 4.8. Oxidative Burst Assay

After surface staining, 500,000 cells were incubated for 15 min at 37 °C in buffer (PBS, 3% FBS, 5mM EDTA) supplemented with 1 μM Dihydrorhodamine (DHR) 123 and 50 μg/mL catalase (Sigma-Aldrich, St Quentin Fallavier, France). Cells were subsequently washed with PSE and the fluorescence intensity of DHR123 was read on a BD LSR Fortessa (BD Biosciences, Le Pont de Claix, France) and analyzed with DIVA V8 and FlowJo V10 software (BD Biosciences, Le Pont de Claix, France).

### 4.9. NADPH/NADP

After surface staining, 500,000 cells were incubated for 30 min at 37 °C in PBS supplemented with a JZL1707 NAD(P)H sensor according to the manufacturer’s instructions (AAT Bioquest, Euromedex, Souffelweyersheim, France). Cells were subsequently washed with PBS and the fluorescence intensity of the JZL1707 NAD(P)H sensor was read in the PE channel on a SP6800 Spectral Cell Analyzer (Sony, Weybridge, UK) and analyzed with FlowJo V10 software (BD Biosciences, Le Pont de Claix, France).

### 4.10. NO (Nitrite) Quantification

First, 10^6^ highly purified neutrophils were cultured for 24 h at 37 °C in DMEM without red phenol (Gibco-ThermoFisher Scientific, Courtaboeuf, France) supplemented with 10% FBS (GE Healthcare, Chicago, IL, USA), penicillin (100 U/mL), and streptomycin (100 μg/mL) (Gibco-ThermoFisher Scientific, Courtaboeuf, France). NO production was measured in the supernatant using Greiss reagent assay according to the manufacturer’s instructions (Sigma Aldrich, St Quentin Fallavier, France). The absorbance was measured at 550 nm using a Multiskan FC plate reader (ThermoFisher Scientific, Courtaboeuf, France).

### 4.11. Arginase Activity Assay

Whole cell lysates of highly purified neutrophils were prepared in 1% Igepal CA-630 (Sigma Aldrich, St Quentin Fallavier, France), 40 mM Tris-HCl, pH 8, 150 mM NaCl, 5 mM EDTA, 5 mM iodoacetamide, 2 mM PMSF, and protease inhibitor mixture (Complete Mini tablet; Roche Applied Scienc, Sigma Aldrich, St Quentin Fallavier, France) at 4 °C for 30 min. The arginase activity level was determined using the Arginase Assay Kit according to the manufacturer’s instructions (Sigma Aldrich, St Quentin Fallavier, France). The absorbance was measured at 450 nm using a Multiskan FC plate reader (ThermoFisher Scientific, Courtaboeuf, France).

### 4.12. TGF-β ELISA

Harvested tumors were immediately frozen in nitrogen liquid and thawed when cell lysates were prepared. A mechanical cell disruption and cell lysis was performed on a FastPrep-24 cell disruptor (MP Biomedical, Illkirch-Graffenstaden, France). Tumors were homogenized in 1% Igepal CA-630 (Sigma-Aldrich, St Quentin Fallavier, France), 10 mM Tris-HCl pH 8, 150mM NaCl, 5mM EDTA, 10% Glycerol, 0.5% anti-foam (silicone), and protease inhibitor mixture (Complete Mini tablet, Roche Applied Science, Sigma Aldrich, St Quentin Fallavier, France) and mixed with 0.5g Lysing Matrix D beads (MP Biomedicals Illkirch-Graffenstaden, France) before mechanical disruption. Supernatants were collected after centrifugation. Total protein quantification was performed using the BCA (bicinchoninic acid assay) protein assay (ThermoFisher Scientific, Courtaboeuf, France) and TGF-β1 concentrations determined by ELISA (Invitrogen, ThermoFisher Scientific, Courtaboeuf, France), according to the manufacturer’s instructions.

### 4.13. Tumor Conditioned Medium and In Vitro Stimulation

Total tumor cells were cultured in complete DMEM medium supplemented with 10% FBS for 24 h. Cells were discarded by centrifugation and tumor-conditioned medium (TCM) was harvested and stored at −20 °C. Aliquots of TCM were depleted of IFN-γ by incubation of TCM with anti-IFN-γ mAb (25 μg/mL) for 1 h at 4 °C followed by adsorption of the complex on Prot G-Sepharose 4B (Sigma-Aldrich, St Quentin Fallavier, France) for 30 min at 4 °C. Control aliquots of TCM were adsorbed on Protein G-Sephapose 4B alone. Treated TCM was collected by centrifugation and stored at −20 °C. Highly purified BM was stimulated in vitro for 24 h with medium alone or in the presence of TCM (1/2) or 10 ng/mL IFN-γ.

### 4.14. Statistical Analysis

Statistical analyses were carried out with Prism software version 6.0 (GraphPad Prism, San Diego, CA, USA). Depending on the data distribution (Shapiro normality test) and matched or not matched observations, a Kruskal–Wallis one-way ANOVA, an unpaired *t* test, or a Mann–Whitney test were used. Pairwise multiple comparisons of experimental groups were performed using two-way ANOVA. *p* ≤ 0.05 was considered statistically significant.

### 4.15. Data Deposition

The data reported in this article were deposited in the Gene Expression Omnibus (GEO) database, https://www.ncbi.nlm.nih.gov/geo (accession nos. GSE133807).

## 5. Conclusions

In conclusion, this extensive in vivo study of TANs infiltrating cSCC underlines their diversity of functions and highlights their predominant involvement in immune suppression of CD8^+^ T cell responses, notably through upregulation of PD-L1. It remains to broaden our knowledge of all the factors and cellular interactions in the TME that modulate the TAN’s phenotype and functions. Targeting of TANs is a promising therapeutic strategy for the treatment of cSCC and other cancers highly infiltrated with protumor neutrophils.

## Figures and Tables

**Figure 1 cancers-12-01860-f001:**
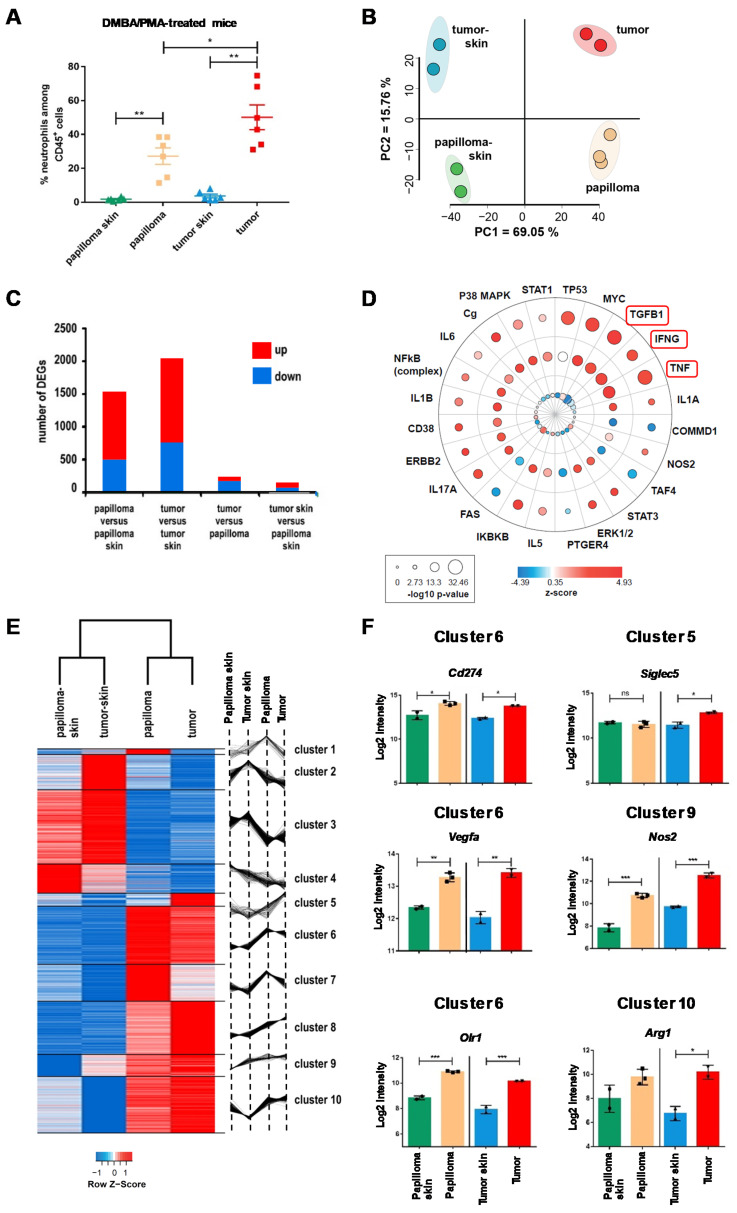
Distinct transcriptional signatures of neutrophils isolated from cSCC lesions and surrounding skin of DMBA/PMA-treated mice highlight their protumor functions. (**A**) Percentage of neutrophils among CD45+ cells in the skin surrounding papillomas (papilloma skin), within papillomas, in the skin surrounding tumors (tumor skin), and within tumors, *n* = 6 mice per group, * *p* < 0.05, ** *p* < 0.01, Mann–Whitney *U* test. (**B**) Microarray analysis of cell-sorted neutrophils: two-dimensional PCA of the 15% most variable probe sets with a minimum log2 average expression of 6. Neutrophils were purified from pools of skin surrounding papillomas (*n* = 6 mice, 2 experiments), skin surrounding tumors (*n* = 9 mice, 2 experiments), papillomas (175 papillomas from *n* = 8 mice, 3 experiments), and tumors (24 tumors from *n* = 5 mice, 2 experiments). (**C**) DEGs in pairwise comparisons. The four contrasts were analyzed based on a minimum log2 average expression of 6, an absolute logFc of at least 1, and an adjusted *p* value *p* ≤ 0.05. (**D**) IPA upstream regulator enrichments from the analysis of DEGs in papilloma versus papilloma skin (outer circle), tumor versus tumor skin (middle circle), and tumor versus papilloma (inner circle). The most significant (−Log10 ≥ 1.39) upstream regulators are shown. A negative z-score (blue) denotes an inhibited pathway. A positive z-score (red) stands for an activated pathway. (**E**) Hierarchical clustering of genes differentially expressed between lesions and their respective skin control identified 10 clusters. The DEGs were selected based on an average log2 expression level across all conditions of at least 7, an absolute log2-fold change ≥ 1, and an adjusted *p* value ≤ 0.05, in at least one of the two contrasts, papilloma versus papilloma skin and tumor versus tumor skin. (**F**) Average intensity expression levels and adjusted *p* values of *Cd274* (PD-L1), *Vegfa*, *Olr1* (LOX1), *Siglec 5* (Siglec F), *Nos2*, and *Arg1* transcripts are shown, * *p* < 0.05, *** *p* < 0.001, limma differential expression analysis. cSCC: cutaneous squamous cell carcinoma; DMBA: 7,12-dimethylbenz[a]anthracene; PMA: phorbol 12-myristate 13-acetate; PCA: principal component analysis; DEGs: differentially expressed genes; IPA: Ingenuity Analysis Pathways; PD-L1: programmed death-ligand 1; LOX1: lectin-type oxidized LDL receptor 1; Siglec F: sialic acid-binding immunoglobulin-type lectins.

**Figure 2 cancers-12-01860-f002:**
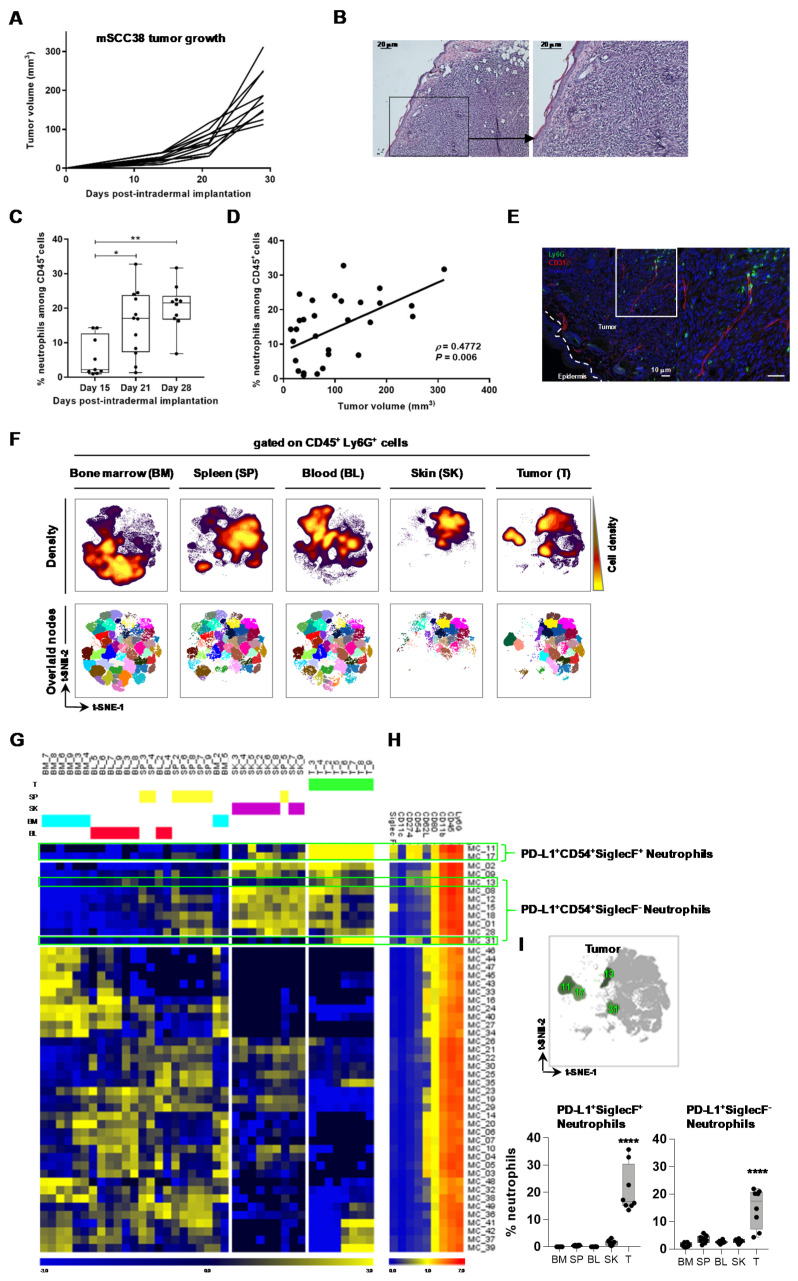
Characterization of TANs (tumor-associated neutrophils) infiltrating mSCC38 tumors. (**A**) Growth kinetic of mSCC38 grafted intradermally in FVB/N mice (*n* = 10 mice). (**B**) Representative hematoxylin-eosin staining of a frozen section, scale bar: 20 μm. (**C**) Proportion of TANs in mSCC38 tumors (*n* = 9–12, from three independent experiments), * *p* < 0.05, ** *p* < 0.01, Kruskal–Wallis one-way ANOVA. (**D**) Correlation between the frequency of infiltrating Ly6G^+^ TANs and tumor volume (*n* = 31). Linear regression curve, spearman r value, and *p* value (95% confidence interval) are shown. (**E**) Representative immunofluorescence imaging of frozen sections of mSCC38 tumor at day 30 post-intradermal implantation showing Ly6G+ neutrophils (green), CD31+ vessels (red), and nuclei (blue). The white dashed line delineates the epidermis–dermis junction; scale bar: 10 μm. (**F**) Neutrophil heterogeneity assessed by FlowSOM automatic clustering after t-SNE (t-distributed stochastic neighbor embedding) dimensional reduction using PD-L1, SiglecF, CD54, CD62L, CD11b, and Ly6G. Cell density for the concatenated file of each group is shown on a black to yellow heat scale. FlowSOM clustering was done to separate neutrophil subsets into 49 meta-clusters (MCs). MCs of merged files of each group were overlaid on a t-SNE map. (**G**) MC proportions heatmap. Samples and mean-centered Log2-transformed MC cell proportion were depicted in a heatmap and arranged according to complete linkage hierarchical clustering. (**H**) MC marker expression heatmap. Markers were arranged according to complete linkage hierarchical clustering, but MCs were ordered according to the (**G**) heatmap MC order. (**I**) Tumor-specific MCs. Four MCs (green gates in (**G**,**H**) were back-viewed on a t-SNE-2/t-SNE-2 map). Cell abundance of PD-L1^+^CD54^+^SiglecF^+^ neutrophil subset (MC 11 and 17) and PD-L1^+^CD54^+^SiglecF^−^ neutrophil subset (MC 31 and 13) is presented as the cell proportion among total neutrophils of each group of samples, **** *p* < 0.0001, Kruskal–Wallis one-way ANOVA.

**Figure 3 cancers-12-01860-f003:**
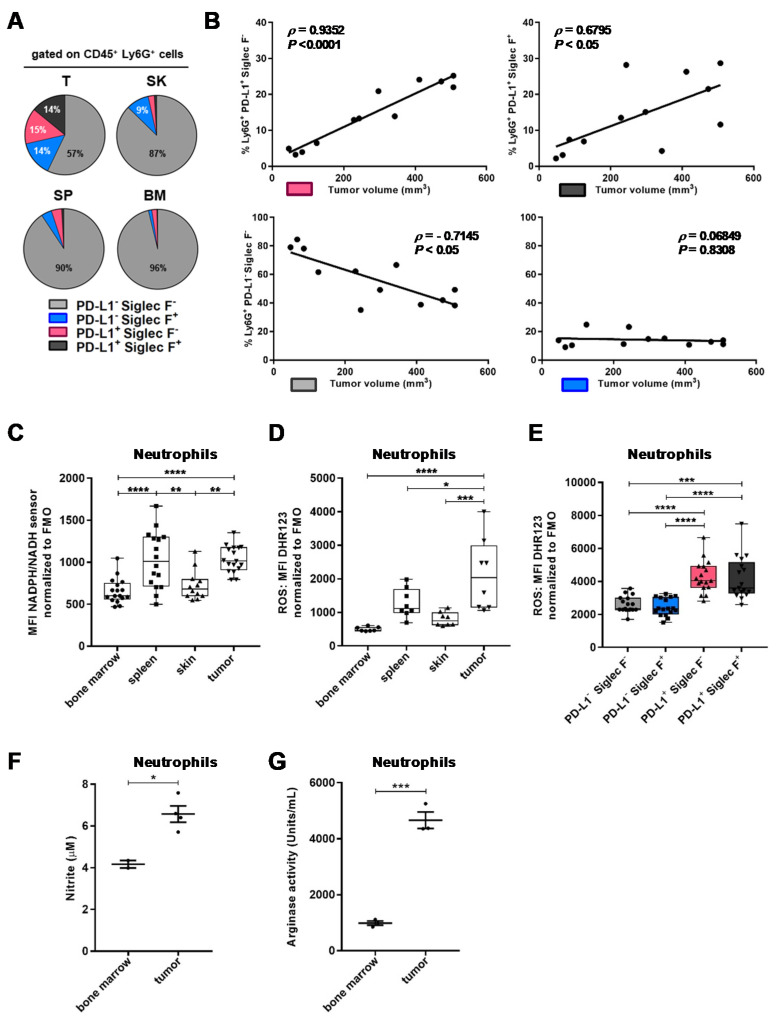
Phenotype and function of TANs infiltrating mSCC38 tumors. (**A**) Proportions of neutrophils expressing either PD-L1 and Siglec F alone or together or none of them, day 30 post-intradermal implantation (*n* = 5–12 mice, mean ± SEM (standard error of mean) from two independent experiments). PD-L1^−^Siglec F^−^ from tumors was significantly downregulated compared to the other subsets, **** *p* < 0.0001; PD-L1^+^Siglec F^−^ from tumors was significantly upregulated compared to skin and BM (bone marrow), * *p* < 0.05; PD-L1^+^Siglec F^+^ from tumors was significantly upregulated compared to skin and spleen, * *p* < 0.05 and to BM, ** *p* < 0.01; PD-L1^−^Siglec F^+^ from tumors was significantly upregulated compared to BM, * *p* < 0.05; Two-way ANOVA. (**B**) Correlation between the frequency of infiltrating PD-L1^+^Siglec F^−^, PD-L1^+^Siglec F^+^, PD-L1^−^Siglec F^−^, and PD-L1^−^Siglec F^+^ neutrophils and tumor size (*n* = 12). Linear regression curve, spearman r value, and *p* value (95% confidence interval) are shown. (**C**) Intracellular NAD(P)H (reduced nicotinamide adenine dinucleotide phosphate) levels and (**D**) ROS (reactive oxygen species) production in neutrophils from the BM, spleen, skin, and tumor from mSCC38-bearing mice (*n* = 8–17 mice, mean ± SEM from two or three independent experiments), * *p* < 0.05, ** *p* < 0.01, *** *p* < 0.001, **** *p* < 0.0001 Kruskal–Wallis one-way ANOVA. (**E**) ROS production in TANs from mSCC38-bearing mice day 30 post-intradermal implantation (*n* = 17 mice, mean ± SEM from three independent experiments), *** *p* < 0.001, **** *p* < 0.0001 One-way ANOVA. (**F**) Nitrite concentrations measured in supernatants from overnight incubation of highly purified neutrophils from mSCC38-bearing mice (*n* = 18 mice, mean ± SEM from three independent experiments), * *p* < 0.05, unpaired two-tailed student’s *t* test. (**G**) Arginase activity measured in whole cell lysates from highly purified neutrophils from mSCC38-bearing mice (*n* = 18 mice, mean ± SEM from three independent experiments), *** *p* < 0.001, unpaired two-tailed student’s *t* test.

**Figure 4 cancers-12-01860-f004:**
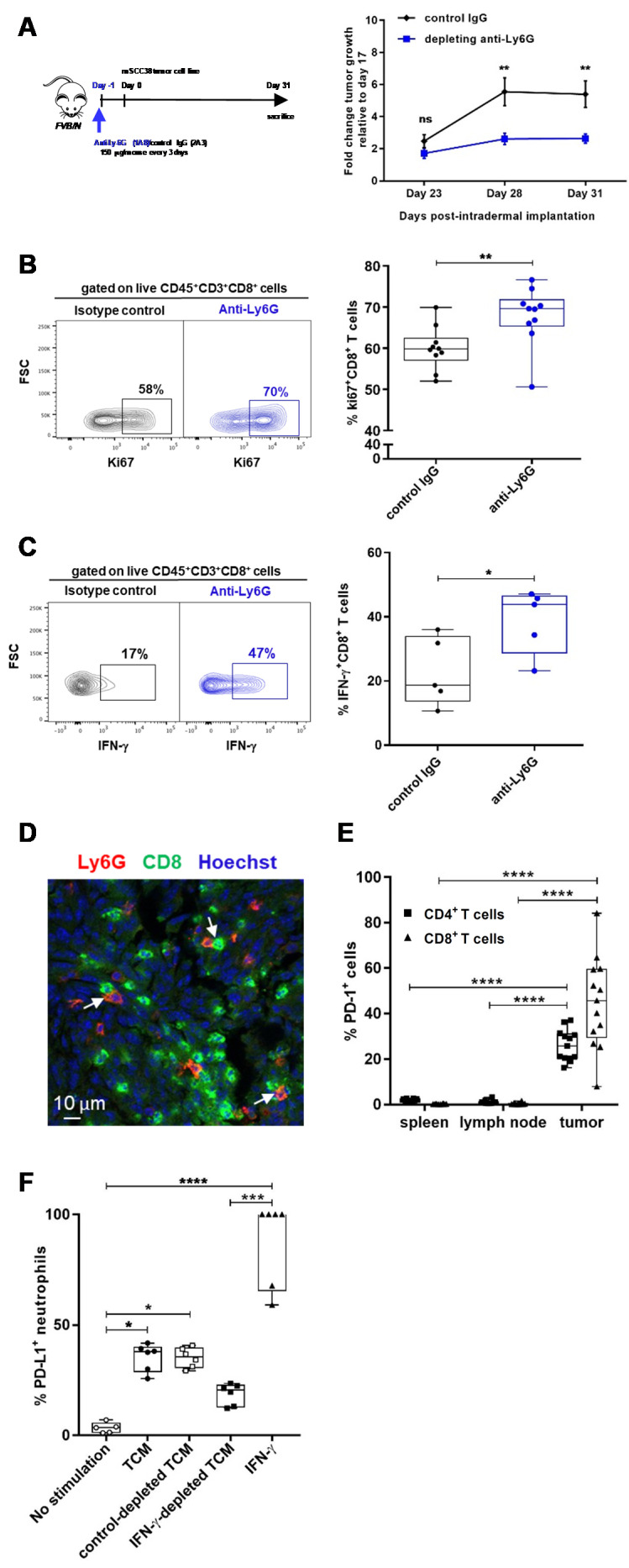
TAN depletion delays cSCC growth and restores antitumor CD8^+^ T cell responses. (**A**) left panel: Experimental protocol of the depletion of neutrophils in mSCC38-bearing mice; right panel: Fold change tumor growth between day 17 and indicated times (*n* = 10 mice per group, mean ± SEM from two independent experiments), ** *p* < 0.01 Two-way ANOVA. (**B**) Frequencies of proliferating Ki67^+^CD3+CD8+ T cells infiltrating mSCC38 tumors in isotype IgG control or anti-Ly6G-treated mice (*n* = 10 mice, mean ± SEM from two independent experiments). ** *p* < 0.01, Mann–Whitney *U*-test. (**C**) Frequencies of IFN-γ-producing CD3^+^CD8^+^ T cells infiltrating mSCC38 tumors in IgG control or anti-Ly6G-treated mice (*n* = 5 mice, mean ± SD). * *p* < 0.05 unpaired student’s *t* test. Representative contour plots are shown (**B**,**C** left panel). (**D**) Representative IF imaging of frozen sections of mSCC38 tumor at day 30 post-intradermal implantation, scale bar: 10 μm. (**E**) Proportions of PD-1+ cells among CD4+ T cells and CD8^+^ T cells from mSCC38-bearing mice day 30 post-intradermal implantation. (*n* = 10–13 mice, mean ± SEM from two independent experiments). **** *p* < 0.0001 Two-way ANOVA. (**F**) Frequencies of PD-L1+ neutrophils following stimulation of cell-sorted BM neutrophils from naive mice with the indicated TCM and cytokines for 24 h. * *p* < 0.05, *** *p* < 0.001, **** *p* < 0.0001 Kruskal–Wallis one-way ANOVA. SD: standard deviation; IF: immunofluorescence; TCM: tumor-conditioned medium.

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
