# Peer review of "Tumor-Associated Neutrophils Dampen Adaptive Immunity and Promote Cutaneous Squamous Cell Carcinoma Development"

_cancers, 2020, doi:10.3390/cancers12071860_

Round 1
Reviewer 1 Report
Well-written manuscript, describing novel mechanistic role of TAN in cSCC.
Reviewer 2 Report
This is a well-written manuscript presenting data from a methodologically well-executed study assessing the the role of tumor infiltrating neutrophils in precancerous and established cutaneous squamous cell carcinoma. The main original results in this study are that tumor associated neutophils in invasive cSCC are heterogeous in term of their phenotype, display protumor gene expression profile and suppress, CD8+ lymphocyte responses through different mechanisms including the induction of increasing PD-1 expression at their cell membrane.
Minor comments:
-Beside PD-1, are increased expression of other checkpoint receptors found on CD8+ T lymphocytes ?
Reviewer 3 Report
Major comment
The study was well-designed and the observations were presented in a well-organized manner. The authors claimed that tumor-associated neutrophils could release nitrates and express arginase to inhibit tumor immunity, but they did not determine the effects of neutrophil depletion on nitrate levels and arginase expression in tumor tissues. The authors should clarify these points clearly.
Minor comments
#1. The authors examined gene expression patterns in mouse tumor tissues. Mice do not possess IL-8/CXCL8 and CXCR1 orthologues. Thus, it is not appropriate to use these gene names to characterize the observed gene expression signatures.
#2. Figure 2E. The quality of the figure is too poor and therefore, should be replaced with a figure of a better quality.
#3. Figure S1E. The authors should define papilloma-3 clearly.
Round 2
Reviewer 3 Report
The authors modified the manuscript mostly in response to the comments.